# Plumage Variation and Sex Ratio in the Brown-Backed Parrotlet (*Touit melanonotus*; Psittacidae)

**Marina Vivianne Carcassola** [1,2,*], **Fernanda Bocalini** [1], **Mercival Roberto Francisco** [3] **and Luís Fábio Silveira** [1]

1   Seção de Aves, Museu de Zoologia da Universidade de São Paulo, São Paulo 04263-000, Brazil; fernanda.bocalini@usp.br (F.B.); lfs@usp.br (L.F.S.)
2   Instituto de Biociências da Universidade de São Paulo, São Paulo 05508-090, Brazil
3   Departamento de Ciências Ambientais, Universidade Federal de São Carlos, Sorocaba 18052-780, Brazil; mercival@ufscar.br
*   Correspondence: marina_vivianne@usp.br

**Abstract:** The Brown-backed Parrotlet, *Touit melanonotus*, is a rare endemic bird to the Brazilian Atlantic Forest, currently considered as "Vulnerable" in the Brazilian Red List of Threatened Species. We estimated the sex ratio of a wild flock of *T. melanonotus* using molecular markers, examined morphological variation in 34 museum specimens to test for sexual dimorphism, and conducted a literature review about sex ratio in Psittacidae for comparative purposes. We found a sex ratio of 0.8:1 (male/female; *n* = 29) in *T. melanonotus*, and a χ2 Goodness-of-fit test showed no significant difference from equality (*p* > 0.05). We describe three main categories in plumage: the first (and most common) comprises uniformly lime green birds, slightly darker on the head. The second is composed of individuals who are overall lighter, with the breast feathers washed with light greenish gray, and feathers of the head being dark lime green, presenting a sharp contrast with the breast feathers. The third and the rarest one is composed of birds with light greenish gray underparts with emerald green and darker upper parts. *T. melanonotus* has no apparent sexual dimorphism. We found no evidence of geographic variation. Sex ratio deviation may not be a parameter increasing the vulnerability of the species. Data like these represent a big leap in the knowledge of the species and has the potential to help and inform conservation efforts.

**Keywords:** parrotlet; conservation; sex dimorphism



## 1. Introduction

The problematic lack of knowledge about Neotropical bird species was recently described as the "Parkerian Shortfall". This gap, which concerns basic natural history data, is an obstacle to understand ecological processes as well as anthropogenic impacts on species and their habitats [1]. One of the less known genera of parrots from the Neotropics—and even of birds in general—is *Touit* Gray, 1855. Data about the natural history are virtually nonexistent for all species of the genus: *Touit melanonotus*, *T. batavicus*, *T. huetii*, *T. costaricensis*, *T. dilectissimus*, *T. purpuratus* (with two subspecies), *T. surdus*, and *T. stictopterus* [2]. Specimens of *Touit* are naturally rare, and phylogenetic relationships within the genus have only recently been investigated [3]. In captivity, birds have never survived for more than a couple of months [4]. Skeletons are also very rare, and the material housed in collections is much inferior to the other genera of birds with similar size and distribution. Considering that three of the eight species are considered "Near Threatened" and one "Vulnerable" by the IUCN [5], it is notable that the lack of data is even more problematic in the light of conservation science, as it hinders the planning and application of conservation strategies.

The Brown-backed Parrotlet *Touit melanonotus* (Wied-Neuwied, 1820) is endemic to the Brazilian Atlantic Forest and occurs patchily in the states of Bahia, Minas Gerais, Espírito Santo, Rio de Janeiro, São Paulo, Paraná and Santa Catarina [4,6–14]. From the little we know about this species, it appears (at least seasonally) from sea level up to montane forests,

about 1700 m [4,6–16], and feeds on fruits [10,12,17–20] and seeds [7], with an apparent preference for *Clusia* spp. fruits, which keep the birds absurdly tame and distracted (LFS, pers. obs.). They do not use their feet to hold the fruits while eating, a behavior also noticed in *T. surdus* [6] and very unusual for Neotropical parrots.

Despite being inconspicuous and silent when perched, making it extremely hard to detect, the Brown-backed Parrotlet is a gregarious species, with groups of 6 to 70 individuals recorded [19]. They are noticed more easily when flying, usually at higher altitudes. They clearly avoid open perches, preferring to feed and rest among the dense foliage of the forest canopy (LFS, pers. obs). The breeding season is likely to occur at the beginning of the rainy season, from September onwards [21], and the generation time is estimated to be at 3.6 years [5]. The species is thought to undertake altitudinal migration, but migratory movements have never been clearly studied and may be more related to the presence of fruiting trees than a predictable, regular displacement between areas of a given population.

The Brown-backed Parrotlet is currently considered as "Vulnerable" in the Brazilian Red List of Threatened Species [16], but curiously, it was recently downgraded to "Near Threatened" by the IUCN [5]. Field estimates indicate that there are less than 10,000 mature individuals in nature, with less than 1000 in each subpopulation, and they face continued population decline. Historical and recent habitat loss are thought to be the causes of the decline, although the exact impacts on the populations are not well known, due to the rarity of the species [16].

*Touit melanonotus* was known only by a handful of records until the 2000s, when a flock was regularly seen and photographed at Ubatuba, on the northern coast of the state of São Paulo, feeding on fruits of *Clusia* spp., a common tree found at that locality. After this discovery, the number of photographs and recordings not only expanded the known distribution but also flooded websites with unpublished and interesting aspects of its natural history and feeding and breeding behavior. Although not systematically collected and opportunistic, these photographs and recordings, upon careful analysis, offer a great opportunity for learning more about this species (see more than 300 photographs taken throughout the species' range on www.wikiaves.com). With a small size, the species follows the genus' standards, measuring about 15 cm in length [21] and being predominantly green in color. Curiously, despite the black back, its English name refers to a color that is not found in these birds. Due to the paucity of material in scientific collections, this species never had a thorough plumage analysis, and its weight and other morphological traits are also poorly known.

Dores et al. [22] presented an interesting report of a flock of *T. melanonotus* received in the Centro de Recuperação de Animais Silvestres do Parque Ecológico do Tietê (hereafter Cras-Pet), a rescue center located in the eastern part of São Paulo city. By the morning of 23 March 2020, the staff received 33 individuals found at Guarulhos International Airport, not far from Cras-Pet. According to the staff of the airport, the parrotlets were found on the ground after colliding with the windows of the airport. Out of the 33 birds found, 29 died and 4 were ringed and released on the next day in the secondary forest at Cras-Pet, with no further information. The 29 birds were submitted to necropsy, which revealed severe wounds and traumas dispersed in the skull, neck, breast, internal organs, and wings, typical of frontal collisions at high speeds (LFS, pers. obs.). Dores et al. [22] found all birds with skulls fully or almost fully ossified (>75% ossification, although a few skulls were damaged and therefore could not be analyzed), developed gonads, and fat dispersed by the carcass. All dead individuals were sexed by gonad observation (13 males, 13 females, and 3 undetermined), weighed, and measured. The specimens were not individually numbered or labeled, and the authors, conscious of the importance of the findings, sent the birds to the Museu de Zoologia da Universidade de São Paulo (MZUSP), where they were prepared as study skins and incorporated into the collection.

This tragic event, however, turned out to be an unprecedented opportunity for a better understanding of many aspects of the biology and natural history of one of the rarest and least known birds in the Neotropical region. In the practical sense, this is one of the few

known examples of an entire flock of adult parrots from the Neotropics being collected at once. The number of specimens in the MZUSP collection, although still modest, can now provide information about plumage variation, for example.

Sex identification is the first step to study sex ratio in a population. Although both random Mendelian sex allocation and the adaptive sex determination model suggested by Fisher [23] predicts a stabilized number of males and females in a population, a review made by Donald [24] supported that skewed adult sex ratios are actually common in wild birds. An asymmetric sex ratio (significantly different than 1:1) is a type of stochastic phenomenon that has the potential to erode the genetic variability of populations, by limiting the number of couples in monogamous birds and consequently reducing the number of individuals generated in the population [25]. In this way, estimating sex ratio is very important to inform conservation actions.

Here, we combined morphology and plumage analysis with molecular sexing to (i) describe three distinct, and not previously reported, intraspecific color categories, (ii) reveal that the plumage aspects previously attributed to sexual dimorphism instead represent individual variation not associated with sex, and (iii) discuss the inconsistencies and discordances between the plumage patterns described by other authors. We also conducted a literature review to find a tendency in Psittacidae sex ratios, for comparative purposes. Data like these not only represent a big leap in the knowledge of *T. melanonotus*, helping to diminish the so-called Parkerian Shortfall, but also have the potential to help and inform conservation efforts.

## 2. Materials and Methods

We examined plumage color variation and patterns in the 34 specimens of *T. melanonotus* housed in MZUSP using Smithe [26] and Munsell [27] to standardize color comparisons and photographs on WikiAves ([www.wikiaves.com](www.wikiaves.com), accessed on 1 February 2023). Of the 307 records on the website, we removed from the analysis those in which we could not clearly observe the plumage, that is, those in which the birds were on their sides, on their backs, or flying too far away. Photos with very dark lighting were also not examined. In cases where there were photos taken on the same day and in the same location, and that most likely portrayed the same individuals, we only reported the photo with the best quality. Thus, we classified a total of 145 photos.

Tissue samples (breast) were collected from the 29 individuals of *T. melanonotus* received by Cras-Pet for molecular sexing. Total genomic DNA was extracted from pectoral muscle tissue samples using the "PureLink®" Genomic DNA extraction kit (Invitrogen, Waltham, MA, USA) following the manufacturer's protocol at the Molecular Biology Laboratory (Multiusers) of the MZUSP. The DNA concentration was quantified with a Qubit fluorometer ("Life Technologies, Inc.", Carlsbad, CA, USA). All individuals were sexed as described in Han et al. [28] via the amplification of homologous copies of the CHD gene (chromo-helicase-DNA binding, or chromium-dependent helicase), located on the Z and W sex chromosomes of birds, using primers P2/P8 [29] and P0 [28] in the same PCR reactions. PCR reactions were modified from Han et al. [28]. A 2 μL DNA aliquot (0.125–0.500 μg DNA) was amplified in 10 μL of PCR reaction mix. Each amplification mix contained $1\times$ Buffer, 1.5 mM $MgCl_2$, 0.2 mM dNTPs, 0.1 μL of Taq Platinum ("Invitrogen Life Technologies", Carlsbad, CA, USA), and 0.35 μM of each primer. An initial denaturation reaction was performed at 94 °C for 2 min 30 s. The amplification reaction was developed using the following thermal profile: 30 cycles of denaturation, 94 °C for 30 s; annealing, 53 °C for 30 s; and extension, 72 °C for 45 s, and a final run at 72 °C for 5 min completed the program. The PCR products were separated via electrophoresis in a 2.5% agarose gel with TBE. The gel was stained with ethidium bromide and photographed under ultraviolet light. Unfortunately, we could not obtain any controls for the individuals who were surgically sexed during the necropsy, considering that the specimens were not individually labeled, and the gonads were removed and not measured at Cras-Pet.

Morphometric parameters (bill length and wing length) were obtained for all the individuals deposited in the MZUSP collection, using a caliper accurate to 0.01 mm. The values obtained were submitted to a Student's t-test to verify whether deviations were sex-biased. We applied the Chi-Square ($\chi 2$) Goodness-of-fit test to investigate if deviations of the expected equality of males and females were significant, using a level of significance of 5%, as in Taylor and Parkin [30].

The literature review on sex ratios focused on Psittacidae, using Google Scholar and the Virtual Library of the Parrot Researchers Group. We included and updated the data provided by Taylor and Parkin [30]. All data were compiled in a table (Supplementary Material Table S1) with variables including age, animal origin, sexing method, numbers of males and females, sex ratio (male/female), level of threat following the IUCN Red List [5], and values regarding the Chi-Square ($\chi 2$) Goodness-of-fit test in order to verify whether deviations were significantly different from equality.

## 3. Results

### 3.1. Molecular Sexing and Plumage Patterns

We found 16 females and 13 males (Figure S1), which means a sex ratio of 0.8:1 (male/female) and a percentage of 44.8% males. The $\chi 2$ test showed no significant difference from equality ($p > 0.05$). It is important to recall that four birds were released and not sexed.

We examined the series of *T. melanonotus* housed at MZUSP (34 specimens) and managed to describe three main categories in plumage (Figure 1A–C). The first category (Figure 1A) was the most common one (with 25 specimens; 10 males, 11 females, and 4 not sexed), where birds were uniformly lime green [26] (reference MZUSP 115246) and slightly darker on the head. The mantle on all birds was uniformly black, such as the tip of the flight feathers. In some specimens (e.g., MZUSP 43776, 115232, 115246, 115251, 115257, 11259), the black on the back did not extend to the rump, as observed in MZUSP 115233 and 115258, and this variation was not associated with either age or sex.

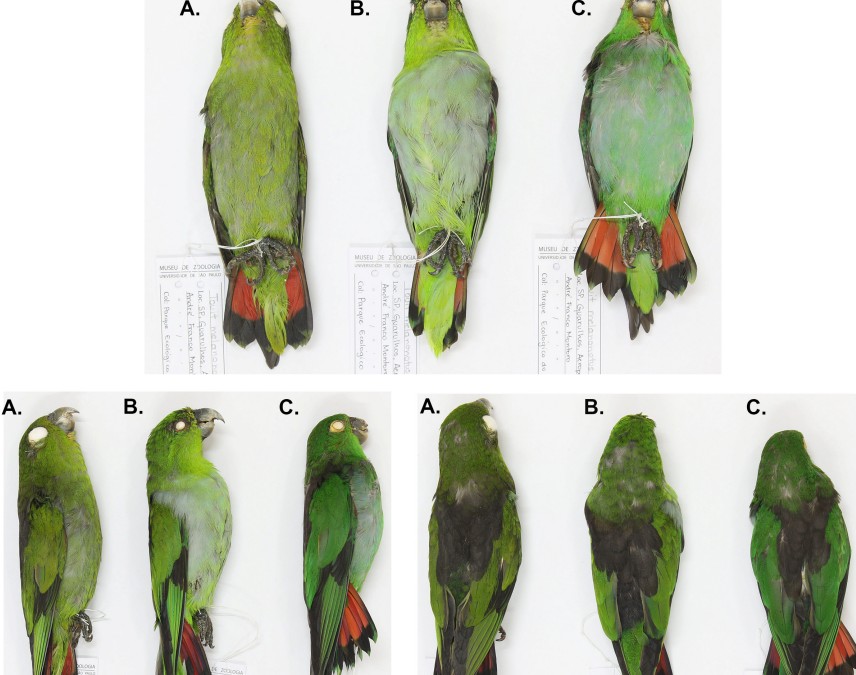

**Figure 1.** Three patterns of plumage observed in *Touit melanonotus* specimens housed at MZUSP. (**A**) The uniformly lime green pattern; (**B**) the overall lighter pattern, with the feathers of the head dark lime green; and (**C**) the emerald green pattern. Photos by Rafael Dantas Lima.

The second category (six specimens; two males, three females, one not sexed; reference MZUSP 115258; Figure 1B) was composed of individuals which were lighter overall, with the breast feathers washed with light greenish gray [27]. The head was dark lime green, with a sharp contrast with the breast feathers. The black of the mantle extended to the rump in all specimens with the lighter pattern. Finally, the third category and the rarest one (with only two specimens, being two females; reference MZUSP 115252; Figure 1C) was composed of birds with light greenish gray and emerald green under parts, and the upper parts were darker than we found in the second group.

In addition, we observed some variation in the tail. The usual pattern was bright green central tail feathers with black bands near to a green tip, and lateral tail feathers with matte red inner webs, also presenting black bands and green tips. However, some individuals lacked the green tips on the tail feathers (MZUSP 115238, 115239, 115240, 115248, 115249, 115251), and this was found more often in males (2:1). Finally, in all specimens, the tail was pointed, unlike the rounded tails found in *T. huetti*, *T. purpuratus*, and *T. surdus* housed at MZUSP.

We could observe all the three patterns described above in the pictures of WikiAves, with a much higher frequency of the first pattern (69%), followed by the second (27.6%) and the third pattern (5.5%). The plumage variation was randomly distributed throughout the range, and in a few cases, different patterns were observed in the same pictures. It is important to recall that the photos published, and therefore the data they provide, can be strongly biased since *T. melanonotus* is a very difficult species to find, which means the photos might not represent all the variation found in the species.

### 3.2. Morphometric Variation

Wing length ranged from 102.2 mm to 117.7 mm (111mm $\pm$ 5 mm) for males, and from 105.3 mm to 118 mm (111.6 mm $\pm$ 3.9 mm) for females. Bill length ranged from 12.64 mm to 14.86 mm (13.60 mm $\pm$ 0.77 mm) for males, and from 12.5 mm to 14.3 mm (13.4 mm $\pm$ 0.6 mm) for females. We found no differences between males and females in both traits ($p > 0.05$) and among the three plumage patterns.

### 3.3. Literature Review

We gathered information from the literature on sex ratios for 24 Psittacidae species, including 10 threatened taxa (4 "Vulnerable", 4 "Endangered", and 2 "Critically Endangered"). From the 39 estimates, 22 were from wild populations (56.4%). Sex ratios varied from 1:1 (total equality) to 8.75:1. We submitted all the studies to the $\chi2$ Goodness-of-fit test and only 5 of the 39 estimates (12.9%) showed significant deviations from equality. From that, three used observational methods to sex individuals and one used captive individuals (in rescue centers), variations that can represent more strong biases. All five of these skewed sex ratios were toward males (Table S1).

## 4. Discussion

The almost equal distribution of males and females in the wild flock of *T. melanonotus* here examined is a result expected for monogamous birds and for Psittacidae species in general, according to our literature review [31–60]. Such data can inform conservation efforts that deviations in sex ratio might not be one of the reasons contributing to the population decline of the species, although more studies like this and preferably on a large space–time scale are required.

Anthropogenic activities, such as poaching, can lead natural populations to present skewed sex ratios. Examples are caged birds for which males are more trapped due to their melodious songs, or game birds for which territorial males are more attracted to decoys or song playbacks used by poachers. Psittacids, on the other hand, often have their offspring stolen from their nests by illegal poachers, in such a way that both male and female young are equally affected. However, in Psittacids trapped at roosting sites, more males might be caught because the females are in the nest incubating or covering the young

at night (Rosemary Low, pers. inf). In the case of the Brown-backed Parrotlet, a bird that has not been maintained in captivity likely due to its highly specialized diet and behavior, trapping or poaching cannot be among the causes of population decline. Furthermore, our data suggest that sex ratio deviation also may not be a parameter increasing the vulnerability of this species, and thorough investigations must be performed to provide a better understanding about the reasons why this is one of the most enigmatic of the Brazilian parrots.

The plumage variation we found is in accordance with that reported in major reference works [21,61], except that we did not observe any markings on ear coverts, as pointed out by Forshaw et al. [61], nor any sexual dimorphism, as Forshaw et al. [61] and Collar et al. [21] found. We interpret the observed plumage variation described herein as individual variation since we did not find any association between plumage variation and sex. We could not attribute the observed variation to age either. Therefore, our findings suggest that *T. melanonotus* has no apparent sexual dimorphism and has at least three plumage patterns. Because the three plumage patterns were present within the same group of individuals derived from the Cras-Pet, and because more than one plumage pattern could be observed in individuals photographed in the same areas, we had no evidence for clinal or interpopulational variation, and the plumage variation has no taxonomic importance.

Studying plumage patterns might clarify phenomena such as sexual selection, social interactions, and environmental factors, including climatic variations [62]. For example, *Forpus xanthopterygius* exhibits brighter and yellowish plumage in drier habitats, while in humid environments, it is duller and darker [63]. Also, according to Collar et al. [64–67], *T. huetii*, *T. dilectissimus*, *T. costaricensis,* and *T. purpuratus* are sexually dimorphic on plumage, most commonly in the coloration of wings and tail feathers. Because a spectrophotometric study found UV phenomena in chromatic plumage in *T. purpuratus* and *T. surdus* [68], individual and/or sexual differences in UV reflections on the plumage might be present, and further research is needed on this topic.

Since, of all the estimates examined on our literature review, only 12.9% showed significant deviations from equality, we may conclude that symmetric sex ratios are most common in Psittacidae than otherwise, which means *T. melanonotus* follows the order tendency. Still, it is important to recall that we applied no corrections to the several differences between the studies, and that we also captured data from studies using captive animals from rescue centers (see Table S1 for more information). Due to the difficulty of analyzing an entire species population and monitoring it over time, studies that assess sex ratios in birds are often biased [24] and differ in many aspects, such as the methodology used, sample sizes, bird ages, and whether sampled animals are from the wild or from captivity. Therefore, it is difficult to draw universal assumptions about a sex ratio tendency in parrots or birds in general without large time–scale computations.

**Supplementary Materials:** The following supporting information can be downloaded at: https://www.mdpi.com/article/10.3390/d15101055/s1, Table S1: Literature review about Psittacidae sex ratio; Figure S1: Results of PCRs performed with 29 individuals of Touit melanonotus.

**Author Contributions:** Conceptualization, L.F.S. and M.V.C.; methodology, all authors; validation, all authors; formal analysis, all authors; investigation, all authors; resources, L.F.S., F.B., M.V.C.; data curation, L.F.S.; writing—original draft preparation, M.V.C.; writing—review and editing, all authors; visualization, all authors.; supervision, L.F.S.; project administration, L.F.S.; funding acquisition, all authors. All authors have read and agreed to the published version of the manuscript.

**Funding:** This research was funded partly due to the ARCA project by São Paulo Research Foundation—FAPESP (2017/23548-2 to L.F.S., and 2022/10252-6 to M.V.C.; 2020/16065-8 to F.B.) and by the National Council for Scientific and Technological Development—CNPq for the Productivity Research Fellowship (Proc# 308702/2019-0, M.R.F., and 302291/2016-6; 308337/2019-0 to L.F.S.).

**Institutional Review Board Statement:** Not applicable.

**Data Availability Statement:** Not applicable.

**Acknowledgments:** Our thanks to Fábio Dores, Lilian Fitorra, Haroldo Furuya, Bruno Simões, Priscila Costa, Liliane Milanelo, and Valéria Pedro, from Cras-Pet, SP, for sending the specimens to MZUSP. We are particularly grateful to J. Battilana for assistance with molecular lab work, Rafael Dantas Lima, Rosemary Low, and the anonymous reviewers for the insightful comments. Nelson Machado Kawall (*in memoriam*) showed the first *Touit melanonotus* to L.F.S. in Ubatuba in the late 1990s, and shared many observations about the behavior of this species. Our great thanks to FAPESP and CNPq for the funding.

**Conflicts of Interest:** The authors declare no conflict of interest. The funders had no role in the design of the study; in the collection, analyses, or interpretation of data; in the writing of the manuscript; or in the decision to publish the results.

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
