# Peer review of "Plumage Variation and Sex Ratio in the Brown-Backed Parrotlet (Touit melanonotus; Psittacidae)"

_diversity, doi:10.3390/d15101055_

Round 1

Reviewer 1 Report

General overview:

Overall, I think the manuscript is well-written and presents some interesting data for a vulnerable species that could have implications for conservation. However, I have a few concerns with the manuscript as it is currently written. First, given that one of the main conclusions of the paper is that there are three plumage morphs not due to sexual dichromatism, I think there needs to be a more quantitative analysis of the morphs. I am fine with the Munsell color chart approach (i.e., as compared to spectrophotometry), but a clustering analysis would be useful showing that the color scores are indeed clumped (e.g., using kmeans). It is hard to assess the support for this conclusion from the photos of 3 specimens shown in Figure 1. Second, although I appreciate the detail in the introduction, I think an additional paragraph attempting to link sex ratios and plumage coloration would be useful. As it stands now, the introduction reads more like a minor advance with a missed opportunity to tie into a broader theme of color polymorphisms and speciation in birds (e.g., Hugall and Stuart-Fox 2012 Nature).

Specific comments:

L15 “ratios”

L22 - Where are the results showing lack of geographic variation?

L57 - add comma after “onwards [21]”

L83 - I think this paragraph could be moved to the Methods.

L94 - “damaged” instead of “injured”?

L115 - The intro is very well written. However, I'd suggest adding a component about color (what's known in Psittacidae, mechanisms, ways people have studied) and how it might tie into sex ratios (e.g., via sexual selection).

L155 - could you explain why you chose these parameters to measure and not others? (tarsus, bill depth)

L175 - I see the differences in the 3 specimens shown in Figure 1. I wonder though, could you provide maybe a supplemental figure of the Munsell scores for each bird to see how much overlap there is between the A-C categories?

L186 - Is there a chance any of the plumage or morphometric variation can be attributed to geography? Lack of geographic variation is alluded to in the abstract, but I don’t see any results/figures showing this.

L212 - Were there any differences in morphometric traits the different plumage morphs?

Table S1 - It would be interesting to see these data plotted as a boxplot or histogram with the value for T. melanonotus highlighted as e.g. a vertical line.

Author Response

We thank the reviewers for the time and effort spent on our research paper. Their comments were extremely valuable and their contributions were fully acknowledged in the revised manuscript. We corrected most of the points addressed by the reviewers, and we replied only to a few of them.

Here we have a point-to-point response to the reviewer’s feedback. Now we firmly believe that our manuscript is greatly improved and represents a new set of information about one of the least-known species of the Neotropics. Our answers are marked in italics and bold just to easy to find them, and a revised version of the manuscript was also uploaded.

Reviewer #1:

Overall, I think the manuscript is well-written and presents some interesting data for a vulnerable species that could have implications for conservation. However, I have a few concerns with the manuscript as it is currently written. First, given that one of the main conclusions of the paper is that there are three plumage morphs not due to sexual dichromatism, I think there needs to be a more quantitative analysis of the morphs. Thank you for the positive income, and we did our best to answer all your concerns. As a result, we have a better manuscript, and your time reading and suggesting were very valuable. The quantitative analysis can be found from line 197 onwards.

I am fine with the Munsell color chart approach (i.e., as compared to spectrophotometry), but a clustering analysis would be useful showing that the color scores are indeed clumped (e.g., using kmeans). It is hard to assess the support for this conclusion from the photos of 3 specimens shown in Figure 1. Second, although I appreciate the detail in the introduction, I think an additional paragraph attempting to link sex ratios and plumage coloration would be useful. As it stands now, the introduction reads more like a minor advance with a missed opportunity to tie into a broader theme of color polymorphisms and speciation in birds (e.g., Hugall and Stuart-Fox 2012 Nature). We appreciate this particular comment, and unfortunately, a spectrophotometer is not available in our museum. As we used both specimens and photos a clustering analysis was problematic. Touit melanonotus is one of the rarest species in the Neotropics, and specimens deposited in scientific collections are extremely rare. Due to the limited number of specimens, we were forced to work with frequencies (see line 197) rather than a more robust statistical analysis. We hope the reviewer can understand that we did the best possible with a very limited sample. We carefully read the paper of Hugall and Stuart-Fox, 2012. This was a very valuable suggestion, but as the authors stated, “The tendency for polymorphic species to speciate more rapidly and to give rise to monomorphic daughter species is predicted by theory”, and “Colour polymorphism may also occur where populations come into secondary contact after diverging in coloration in allopatry or when colour forms are under disruptive selection associated with different microhabitats. In both cases, colour polymorphism may represent incomplete speciation (transient polymorphism) and under certain conditions (for example, when accompanied by selection against hybrids and/or assortative mating) the speciation process will culminate in the evolution of monomorphic daughter species. In sum, a combination of ecological, geographic and genetic processes may accelerate speciation in colour-polymorphic species; our data provide empirical confirmation of a link between colour polymorphism and relatively rapid, recent speciation. Polymorphism in coloration is often associated with differences in suites of correlated traits (for example, morphology, physiology, behaviour and life history) due to correlational or epistatic selection or shared developmental pathways”. Unfortunately, our data cannot provide any advance on this topic, which is out of scope of the present manuscript.

Specific comments:

L15 “ratios”: corrected

L22 - Where are the results showing lack of geographic variation? The results are shown in lines 199-201.

L57 - add comma after “onwards [21]”: corrected.

L83 - I think this paragraph could be moved to the Methods. We respectfully ask the reviewer to keep this information at this point – here we present how the birds arrived at MZUSP, with little information available. We respectfully request this paragraph to be kept in the Introduction.

L94 - “damaged” instead of “injured”? Thank you, corrected.

L115 - The intro is very well written. However, I'd suggest adding a component about color (what's known in Psittacidae, mechanisms, ways people have studied) and how it might tie into sex ratios (e.g., via sexual selection). Done.

L155 - could you explain why you chose these parameters to measure and not others? (tarsus, bill depth): Thanks for calling our attention to this point – as explained in the Introduction, many bills were damaged due to the shock with the window, hindering a proper comparison with statistical support.

L175 - I see the differences in the 3 specimens shown in Figure 1. I wonder though, could you provide maybe a supplemental figure of the Munsell scores for each bird to see how much overlap there is between the A-C categories? Thank you for seeing the differences among the three plumages found in Fig. 1. However, there are no overlap between them, and as we show in the lines 197-200 and in the discussion.

L186 - Is there a chance any of the plumage or morphometric variation can be attributed to geography? Lack of geographic variation is alluded to in the abstract, but I don’t see any results/figures showing this. No, there is no chance, and this was one of the points we paid special attention to. The main reason to refute this hypothesis (which is extremely important, and we thank the reviewer for pointing this out) is that the three patterns we found are present in the same flock (see Fig. 1). The largest series of Touit melanonotus in the world is housed at MZUSP, which also has the broadest geographic distribution of this species. Initially, we planned to include a map showing the distribution of the plumage morphs, but the number of specimens deposited in the museums is very low (out of MZUSP, only more five) and the map will bring no additional information.

L212 - Were there any differences in morphometric traits the different plumage morphs? There are no morphometric differences among the three plumage morphs. This information was added to the manuscript, and we thank the reviewer for calling our attention to this important point which was missing.

                We thank you for your valuable support and suggestions. We expect to have clarified our points and we modified the manuscript accordingly. Please do not hesitate to ask any further questions regarding this manuscript.

Reviewer 2 Report

I have a few suggestions that may improve the quality of this work.

1) The sample size is small, especially from a single flock for the species.  I searched a few other museum inventories and it appears there are other T. melanonotus specimens available elsewhere.  It would be helpful to expand this study, also given the geographic interest of these authors.

2) Color measurement was weak here.  There are more rigorous spectral or photographic procedures available for scoring bird colors - these should be used here and would permit robust testing of sexual dichromatism.

3) Just two morphometric measures were taken (bill length and wing length), which seems not to be making the most of the specimens and literature.  Several bill parameters could be measured, as well as aspects of skull, tarsus, etc.  A more comprehensive morphometric evaluation would make this a stronger paper, including added analyses of sexual size dimorphism.

4) I did not think the Literature Review part of the Results fit into this work.  At most this is a commentary for the Discussion - and it has many weaknesses, including those mentioned by the authors as well as the small sample size of 29 in this study.

none

Author Response

We thank the reviewers for the time and effort spent on our research paper. Their comments were extremely valuable and their contributions were fully acknowledged in the revised manuscript. We corrected most of the points addressed by the reviewers, and we replied only to a few of them.

Here we have a point-to-point response to the reviewer’s feedback. Now we firmly believe that our manuscript is greatly improved and represents a new set of information about one of the least-known species of the Neotropics. Our answers are marked in italics and bold just to easy to find them, and a revised version of the manuscript was also uploaded.

Reviewer #2:

I have a few suggestions that may improve the quality of this work.

1) The sample size is small, especially from a single flock for the species. I searched a few other museum inventories and it appears there are other T. melanonotus specimens available elsewhere. It would be helpful to expand this study, also given the geographic interest of these authors. Thanks for your input and time reading and commenting on our manuscript. Yes, the reviewer is right about the sample size, which is really small. On the other hand, this is the largest series analyzed in history so far. Additionally, we are working with one of the rarest species of the Neotropics in scientific collections. The largest series of Touit melanonotus in the world is housed at MZUSP, which also has the broadest geographic distribution of this species. The number of specimens deposited in other Brazilian museums is very low (out of MZUSP, only five more, three of them with doubtful origin and very old, dating from the early 20th century). According to Collar et al. 1992 (Threatened Birds of the Americas) a handful of old specimens, also with generic origin (e. g. Brazil) are found in foreign museums. The hypothesis that the plumage variation could be related to geography was refused by finding the three plumage patterns in the same flock, from the same locality. This is an important finding of the paper because the variation found in the plumage cannot be related to any geographical pattern.

2) Color measurement was weak here. There are more rigorous spectral or photographic procedures available for scoring bird colors - these should be used here and would permit robust testing of sexual dichromatism. We appreciate this particular comment, and we recognize the limitation of our analysis on this subject but, unfortunately, a spectrophotometer is not available in our museum. We hope the reviewer can understand this limitation.

3) Just two morphometric measures were taken (bill length and wing length), which seems not to be making the most of the specimens and literature. Several bill parameters could be measured, as well as aspects of skull, tarsus, etc. A more comprehensive morphometric evaluation would make this a stronger paper, including added analyses of sexual size dimorphism. We agree with that, but most of our specimens had some traits such as the bill severely damaged due to the shock with the window, hindering a proper comparison with statistical support. We were able to obtain good morphological data only from these structures.

4) I did not think the Literature Review part of the Results fit into this work. At most this is a commentary for the Discussion - and it has many weaknesses, including those mentioned by the authors as well as the small sample size of 29 in this study. We respectfully request the reviewer to reconsider this decision. The information about this topic (plumage variation in parrots) is scattered and still poorly studied. Although our manuscript deals with a broader subject, finding three distinct plumage patterns lead us to highlight this topic and call the attention of other researcher for this frequently neglected issue. Finally, although our sample size is small, we must point out that this is the largest series of this species analyzed so far, as it is one the rarest species in the Neotropics. The number of specimens deposited in scientific collections of the world, collecting since 1820 and including those old, with doubtful or no locality, is inferior to 50.

                We thank you for your valuable support and suggestions. We expect to have clarified our points and we modified the manuscript accordingly. Please do not hesitate to ask any further questions regarding this manuscript.
